# Liver Damage Is Related to the Degree of Being Underweight in Anorexia Nervosa and Improves Rapidly with Weight Gain

**DOI:** 10.3390/nu14122378

**Published:** 2022-06-08

**Authors:** Ulrich Cuntz, Ulrich Voderholzer

**Affiliations:** 1Schoen Clinic Roseneck, Am Roseneck 6, 83209 Prien am Chiemsee, Germany; uvoderholzer@schoen-klinik.de; 2Forschungsprogramm für Psychotherapieevaluation im komplexen Therapiesetting, PMU Paracelsus Medical University Salzburg, 5020 Salzburg, Austria; 3Department of Psychiatry and Psychotherapy, Ludwig-Maximilians-Universität München (LMU), 80336 Munich, Germany

**Keywords:** anorexia nervosa, transaminases, liver damage, autophagy, weight gain

## Abstract

Background: The present study investigates the relationship between hypertransaminasemia and malnutrition on the basis of a very large number of patients. We assume that the level of transaminases not only reflects the extent of underlying liver cell damage but also provides information about the metabolic situation under conditions of energy deficiency. Methods: We present an observational study in two different samples. The first sample consists of 3755 patients (mean age 22.7 years, Range 12–73 years; mean BMI 15.4 kg/m^2^, range 8.1–25.7) out of a total of 4212 patients with anorexia nervosa treated in the Roseneck Clinic within five years for whom a complete admission laboratory was available. The second sample was obtained from a special ward for medically at-risk patients with eating disorders. During the period in question, four hundred and ten patients with anorexia nervosa were treated. One hundred and forty-two female patients (mean age 26.4 years, Range 18–63 years; mean BMI 11.5 kg/m^2^, range 8.4–13) had a BMI of thirteen or less and a complete data set was obtained at admission and weekly in the following four weeks after admission. Results: The increase in liver transaminases shows a very high correlation with weight in sample one (N = 3755). The analysis of variance shows highly significant (<0.001) correlations with an F-value of 55 for GOT/AST and 63 for GPT/ALT. Nevertheless, the variance within the groups with the same BMI is quite high. With re-nutrition in sample two, GOT/AST decreased on average from 71 U/L to 26 U/L (MANOVA F 10.7, *p* < 0.001) and GPT/ALT from 88 to 41 U/L (F = 9.9, *p* < 0.001) within four weeks. Discussion: Below a BMI of about 13, the nutritional status of the patients becomes so critical that the energy supply of the patient is increasingly dependent on the autophagy of the liver, which can be seen in the very strong increase in transaminases here. Refeeding leads very quickly to the normalisation of the transaminases and, thus, a stabilisation of the metabolism leading also to a decrease in autophagy.

## 1. Introduction

For many years, there have been numerous reports in the scientific literature about elevated transaminases in the context of anorexia nervosa (AN) [1,2,3,4,5,6,7,8,9,10,11,12,13,14,15,16,17,18,19,20]. These are mostly case reports. In some reports, the term ‘transaminitis’ is used [1,3,21], but this is not associated with any pathogenetic concept. With the term ‘hepatitis’, which has been used by some other authors [12,17,22], a definition is made at least of what is to be expected in histology: histologically, hepatitis is characterised by single-cell necrosis. There is oedema of the portal fields and a portal infiltrate by lymphocytes, histiocytes and a few plasma cells. However, there are no liver histologies in anorexia nervosa that prove hepatitis in this narrower sense.

In cases of extreme underweight in the context of anorexia nervosa, vital-threatening liver failure is a frequent complication [8,12,14,23,24]. It is therefore of great importance to better understand the patho-mechanism of liver cell damage, as well as how and when transaminases rise and in which cases liver failure is imminent.

The group of authors led by Pierre-Emanuel Rautou [10,25] performed liver biopsies for the first time in a series of twelve patients with AN. In these patients, the liver biopsies were performed on the 1st to the 9th day of the stay, always when the highest value of AST (GOT) was measured. All these patients had a very low weight with a mean BMI of 11.3. Not a single one of these patients had any evidence of other chronic liver disease. All patients had incipient liver failure, defined by a drop in prothrombin to less than 50% of normal levels or an increase in INR above 1.7.

Histology showed no portal fibrosis or periportal inflammatory infiltrates in any of the twelve patients. Glycogen stores were obviously largely depleted in all patients. The most important finding was revealed by electron microscopy: autophagosomes could be detected in many of the biopsies. Autophagosomes are vesicular intracellular compartments containing sequestered material from cell organelles such as mitochondria or the endoplasmic reticulum. The sonography of the liver of these patients showed a normal echogenicity pattern without abnormalities in each case. The authors come to the conclusion that liver cell damage is present and, thus, the increased transaminases in all twelve patients can ultimately be explained neither by liver cell necrosis nor by apoptosis but by the mechanism of autophagy. 

Since liver biopsy usually has no therapeutic consequences in the treatment of severely underweight patients with AN, there are few case reports of liver biopsies and electron microscopic histologies in the literature. Restellini et al. [26] report on another case of a patient with a BMI of 14 and transaminases each significantly above 1000 U/L. The findings confirm those of Rautou et al.: the electron microscopic examination showed clearly reduced glycogen stores, and the number of mitochondria and the rough endoplasmic reticulum were reduced. In contrast, there was an increased number of Golgi bodies and primary and secondary lysosomes corresponding to autophagosomes containing digested cellular material. 

The concept of macroautophagy, i.e., the self-dissolution and self-digestion of cell structures, was introduced as early as the 1960s on the basis of the electron microscopic discovery of lysosomes in rat liver [27]. In the 50 years since then, studies on liver autophagy have yielded many important findings. The liver is rich in lysosomes and has a high level of metabolic autophagy, which is precisely controlled by the concentration of hormones and amino acids. Autophagy in the liver supplies starved cells with amino acids, glucose and free fatty acids for energy production and synthesis of new macromolecules and also controls the quality and quantity of organelles such as mitochondria. Autophagy is a mechanism for energy provision that becomes increasingly important during periods of energy deficiency [28]. There is growing evidence that autophagy in the liver contributes to basic liver functions such as glycogenolysis, gluconeogenesis and β-oxidation, through the selective turnover of specific cell contents, which are hormonally controlled by a number of transcription factors.

There are three defined types of autophagy: macro-autophagy, micro-autophagy and chaperone-mediated autophagy, all of which promote proteolytic degradation of cytosolic components at the lysosome. In the case of AN or in the case of high-grade energy deficiency, it is obviously a matter of macro-autophagy according to the above-mentioned findings. Here, the cytoplasmic charge is transported to the lysosome via a vesicle with a double membrane, the so-called autophagosome, where it fuses with the lysosome and forms an autolysosome [29]. 

For the present study, however, the focus is not on the histological findings in the narrow sense, but rather on the mechanisms underlying the increase in transaminases. Autophagy maintains the health of cells and tissues by replacing obsolete and damaged cellular components with new ones. In the case of energy deficiency in the context of malnutrition, it provides an internal source of nutrients for energy production and thus for the survival of the organism [30]. We must assume that in these cases autophagy leads to limited cell death and thus to an increase in liver enzymes.

Gluconeogenesis and ketogenesis are critical for maintaining energy homeostasis and physiological activity during starvation in mammals and are coordinately regulated in different tissues. The liver is essential for both ketogenesis and gluconeogenesis, and autophagy appears to be primarily a requirement for the provision of ketones [31,32]. 

In line with the assumptions of Kheloufi et al. [25], we hypothesise that starvation-induced autophagy plays a dual role in AN. During the first phase of weight loss, abnormalities in liver blood tests remain moderate, suggesting that autophagy can cope with nutrient deficiency. During this time, autophagy acts protectively and prevents cell death. If starvation continues and BMI reaches a critical level of 13 kg/m^2^ or less, excessive activation of autophagy leads to hepatocyte cell death and liver failure.

The present study investigates the relationship between hypertransaminasemia and malnutrition based on the results from a very large number of patients. The increase in liver transaminases is usually due to cell death, by which the enzymes in the cytoplasm can pass into the blood. This may be due to hepatocellular necrosis, hepatic inflammation, or the autophagy process described above. 

According to previous findings, autolysis of liver cells is based on malnutrition, which forces the organism to provide energy from the cell organelles themselves. In this case, the level of transaminases should correlate highly with BMI, which is (although not in perfect form) a measure of the available energy in the body. 

GOT is bound in the liver cell, especially in the mitochondria. In liver cell necrosis, the quotient of GOT/GPT, the so-called de Ritis quotient, therefore increases. The electron microscope images of liver cells in AN known so far, however, show a depletion of mitochondria, so that in this case the de Ritis quotient could even decrease. 

Upon refeeding, the energy stores are quickly replenished so that a higher degree of ketogenesis is no longer necessary. In this respect, it can be assumed that the transaminases will decrease rapidly, which should be less the case in the case of inflammatory or necrotic damage to the liver.

## 2. Materials and Methods

### 2.1. Participants

All participants had been inpatients admitted to Schön Klinik Roseneck, a hospital specialized in the treatment of mental disorders, which offers 15 specialized wards for patients with eating disorders (Table 1). 

Sample 1: Between 1.5.2015 and 31.8.202,1 4121 AN inpatients have been admitted for treatment in this clinic (some of them had been in remission). For 3755 inpatients a complete data set was available (admission data including weight, age, BMI and relevant laboratory parameters for further analysis (mean age 22.7 years, Range 12–73 y; mean BMI 15.4 kg/m^2^, range 8.1–25.7). This sample also includes older patients. 27 (0.7%) were 60 years old or older. We only used discharge diagnoses, so we can assume that other physical causes of underweight were excluded.

In order to compare body weight and BMI between adolescents and adults we used z-transformation based on LMS-parameters derived from the data of the Center for disease control, USA [33]. For all adults we used LMS-parameter for 240 months of age (see Table 2).

The patients from Sample 2 (*n* = 520) were admitted in the same period of time to an intermediate care unit specialised in the treatment of severely ill ED patients. Most of these patients had been transmitted from other hospitals because of medical problems. The ward offers medical treatment comparable to an intermediate care unit with (whenever needed) permanent monitoring of cardiovascular parameters, frequent controls of laboratory parameters and availability of ultrasound, body impedance analysis, ECG and echocardiography. During this four-year period, 410 out of the 520 admitted patients had an AN primary diagnosis.

In order to provide a sufficient observation period for potential refeeding syndrome occurrence, only those inpatients were included who stayed at least 28 days. Standard treatment of these severely malnourished inpatients comprised at least one weekly blood test, initial monitoring of cardiovascular parameters and weekly multi-frequency body impedance measurements. Phosphate (1024 mg/day) and thiamine (200 mg/day) were supplemented routinely. Although weekly measurements are part of standard procedures, the complete data set was not always available. One hundred and forty-two inpatients with a BMI < 13 had a data set complete enough to be included (no missing data at admission and after 28 days and at most one missing in between) (see Table 1).

The present study used only retrospective data in anonymised form. According to the guidelines by the institutional review board of the LMU Munich, retrospective studies conducted on already available; anonymised data are exempt from requiring ethics approval.

### 2.2. Renutrition

All patients receive three meals with an average total energy content of 2000 kcal per day from day 1 after admission to the ward, divided into three main meals with a choice between vegetarian and non-vegetarian meals. The caloric intake is adjusted and increased according to weight development to aim for an increase in body weight of 700–1000 g/week. The criterion for a sufficient food intake is weight gain, which should be at least 100 g per day. If the weight gain cannot be achieved, the portion size of one or more main meals is increased and up to three snacks between meals are added. In addition, liquid food is offered to substitute for energy losses in case of incompletely eaten meals. All meals are therapeutically accompanied by a nurse or therapist in a 1:6 group supervision. Patient adherence to dietary intake is supported through daily therapeutic contacts and medical rounds. Patients eat their meals in a stable group setting and support each other. Peer pressure may play an important role. In weekly eating protocol sessions patients review their progress and commit to new goals related to hitherto avoided food, fears and counteractive behaviour. Patients do not receive nasogastric feeding since the normalisation of eating behaviour is a general therapeutic goal. The average caloric value (data provided by the caterer and controlled in samples) for the non-vegetarian menu is 743, 717 and 704 (total 2164) kcal for breakfast, lunch and dinner; and 743, 737 and 683 (total: 2162 kcal) for the vegetarian option. If patients do not finish their meal (50–99% eaten) they are asked to drink 1 supplemental drink (400 kcal, Fresubin^®^ 200 mL with 2 kcal/mL). If they eat less than 50% of their respective meal, they are asked to replace the missed-out calories by drinking 2 supplemental drinks (2 × 400 kcal). The group setting and the support from experienced therapists are considered essential for compliance with dietary adherence. Contingency measures are often necessary in order to regulate excessive exercising and slow weight gain. During the first 28 days, it is rare that patients require more than 2000 kcal/day but many patients receive liquid food in order to replace unfinished meals. Patients reduce their calorie consumption by limiting their physical activity. Contingency contracts and video surveillance with 24/7 nursing staff presence as well as very moderate exercise therapy play an important role in normalizing exercise behaviour.

In individual cases, an energy intake of more than 4000 kcal is necessary to ensure sufficient weight gain. All patients participated in an intensive therapeutic program adapted to the special needs of the severely underweight women and suitable for therapeutically addressing the considerable anxieties and resistance associated with weight gain. 

The 142 patients reliably gained weight during the therapy program. The mean BMI increased from 11.4 kg/m^2^ to 13.1 kg/m^2^. The mean weight gain was 4.4 kg, i.e., approx. 1.1 kg per week. The mean weight gain was highest in the first week with approx. 2.1 kg. A standard deviation of 2.2 kg for the entire four weeks shows that there were large interindividual differences in weight gain.

### 2.3. Statistical Analysis

Interrelationships between laboratory parameters and BMI were calculated as Pearson correlation. 

We used One-way-Analysis of variance (ANOVA) in order to test the relationship between BMI-classes and laboratory parameters, respectively the Friedman-Test for the relationship between score and BMI-classes. 

Multiple measurements analysis of variance (MANOVA) was calculated to analyse the change of parameters within the process of weight gain. 

## 3. Results

The increase in liver transaminases shows a very high correlation with weight in sample 1 (N = 3755) (see Figure 1). The analysis of variance shows highly significant (<0.001) correlations with an F-value of 55 for GOT and 63 for GPT. Nevertheless, the variance within the groups with the same BMI is quite high. The Pearson correlation is highly significant at 0.25, but not very high, so that the variance explanation for the increase in transaminases by BMI alone is only moderate. 

The de Ritis quotient (quotient of GOT/GPT) is significantly lower in those patients with lower BMI, i.e., the increase in GPT with lower weight is higher in relation to GOT. 

A comparison of the age groups shows that both GOT and GPT are related to the age-normalised BMI to the same extent.There are no age-specific differences here. The de Ritis quotient appears to be higher in younger patients in the graph. However, the MANOVA does not show any significant covariance for the age group (see Figure 2).

The 142 patients who participated in the eating disorder therapy in intermediate care gained weight rapidly, especially in the first week (most likely due to hypovolaemia on admission). The average weight gain in the first week was 2.1 kg, followed by at least 700 g in the following three weeks (see Figure 3), demonstrating the efficiency of the therapy programme. The energy intake, starting at 2000 kcal, was already initially sufficient to allow a rapid normalisation of the energy balance. 

More than half of all these severely underweight patients in this sample had elevated transaminases. During the course of weight gain, GOT/AST decreased on average from 71 U/L to 26 U/L (MANOVA F 10.7, *p* < 0.001) and GPT/ALT from 88 to 41 U/L within four weeks (F = 9.9, *p* < 0.001). The de Ritis quotient decreased highly significantly from an average of 1.08 to 0.8 (*t*-test *p* < 0.001), i.e., the GPT normalised faster than the GOT.

## 4. Discussion

The results of this largest study to date on the subject of transaminase elevation in AN are quite clear and unequivocal. The lower the weight, the higher the transaminases. There are no significant differences in age. The steepest increase occurs below a BMI of 13 kg/m^2^ or a Z-value of −5. With a high-calorie diet at the beginning, the values largely normalise within four weeks. In individual cases, a further initial increase in transaminases is observed—in no single case did this lead to liver failure. On the other hand, in everyday clinical practice, patients with high transaminases are at high risk of possible liver failure with high mortality risk. 

The result is consistent with the findings of the two smaller studies by Hanachi et al. [24] and Rosen [20]. Except for these two studies, there have been no studies with larger case numbers on the topic of transaminase elevation and liver cell damage in AN. Liver cell damage in the context of being underweight plays a major role in the risk spectrum of AN. In clinical practice, the transition from elevated transaminases to liver failure is observed more frequently and then leads to very critical conditions that require intensive medical treatment. On the basis of the electron microscope examinations of the liver parenchyma to be found in the case of elevated transaminases and high-grade anorexia nervosa, we must assume that the liver cells exhibit an increased degree of autophagy and thus suffer increasing damage.

In animal models, starvation and malnutrition in the liver lead to the occurrence of autophagy of the liver cells. Autophagy is an adaptive mechanism that maintains the energy supply of the organism in a deficient state. The data that are available for this study support this hypothesis: the increase in transaminases is higher the more critical the organism’s supply situation is based on the BMI. On the other hand, the transaminases (and thus, presumably, the underlying mechanism of autophagy) drop very quickly when the patient’s energy supply is restored. In necrotic liver changes, the de Ritis quotient usually increases with increasing damage. In our patients, an increasingly low de Ritis quotient is found with higher degrees of being underweight. Even if this quotient has a rather low significance in the diagnosis of liver diseases, it could be explained by the lack of mitochondria in the hepatocytes, which is conspicuous by electron microscopy. In contrast to GPT/ALT, GOT/AST is predominantly located in the mitochondria.

When interpreting the usual routinely collected laboratory diagnostics, the transaminases stand out. They correlate most strongly with BMI [34,35]. If the presumed mechanism of liver cell autophagy is taken as a basis, the transaminases are particularly suitable for estimating the energy supply of the patient’s organism. Only if no more externally supplied energy is available, must recourse be made to the cell structures available in the liver. The higher the transaminases, the more urgent the external energy supply appears to be. However, we cannot completely exclude the possibility that, in individual cases, other reasons were responsible for the increase in transaminases, such as medication side effects, isolated cases of chronic hepatitis or even malignant diseases. On the other hand, these cannot be responsible for the high correlation between body weight and transaminases.

## 5. Conclusions

In the treatment of AN, the assessment of the risk present in the individual case is of great practical importance from which the need for treatment and, if necessary, emergency measures, are derived. AN is the disease of adolescence and young adulthood with the highest mortality rate [36,37,38]. However, it is still unclear what causes the high mortality. Even in this very large sample, there are many cases that have almost normal laboratory values despite being extremely underweight, i.e., they are obviously well adapted to the very low weight. BMI is a measure introduced to relate body weight to height and thus to predict the morbidity and mortality risk of obesity. Whether it is also a suitable measure for determining the risk in AN has not yet been tested, which is also due to the fact that it is not clear enough what the ‘morbidity’ in AN actually is, i.e., which factors cause the high mortality. In this sense, the changes in the liver could be a good criterion for validating corresponding weight-to-body size ratios. This is all the more true when the metabolic processes underlying the liver changes are better understood.

## Figures and Tables

**Figure 1 nutrients-14-02378-f001:**
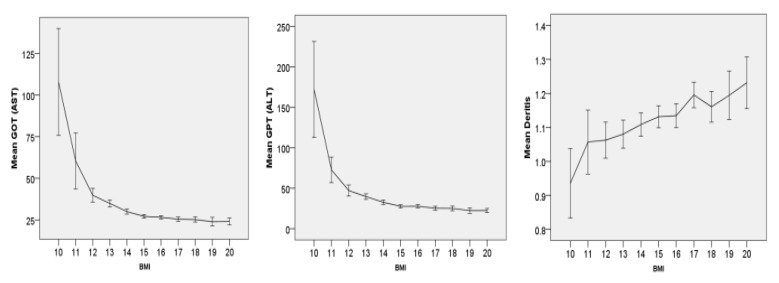
N = 3755—Mean values for GOT (AST), GPT (ALT) and the de Ritis quotient (GOT/GPT) as a function of BMI (integer)—The relationship in the analysis of variance of GOT (F = 55.7), GPT (63.8) and de Ritis (F0 5.7) quotients is highly significant *p* < 0.001 in each case. Error bars represent 95% confidence interval.

**Figure 2 nutrients-14-02378-f002:**
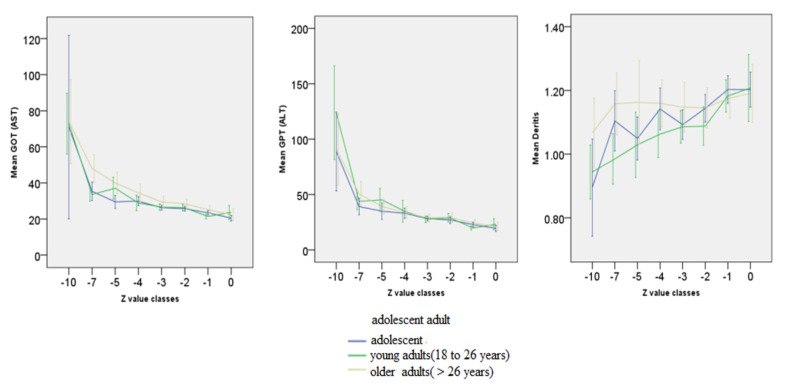
Age-specific N see Table 2—Relationship between BMI and GOT (AST), GPT (ALT) and de Ritis quotient (GOT/GPT) depending on age. Adolescent < 18 y; young adults 18 to 16 y; older adult > 26 y. Z-transformation of BMI using the CDC’s LMS table. Error bars represent 95% confidence interval.

**Figure 3 nutrients-14-02378-f003:**
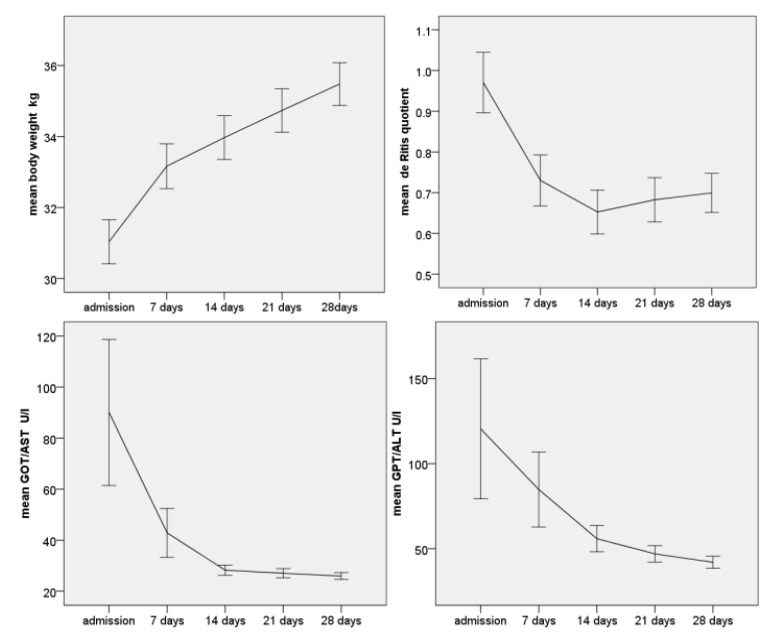
N = 142 Change in body weight in kg (MANOVA *p* < 0.001, F = 347.0), GOT (MANOVA *p* < 0.001, F = 17.5), GPT (MANOVA *p* < 0.001, F = 13.2) and de Ritis quotient (MANOVA *p* < 0.001, F = 54.9) during the course of refeeding. Error bars represent 95% confidence interval.

**Table 1 nutrients-14-02378-t001:** Sample characteristics.

	Sample 1	Sample 2
	All Wards for ED	Intermediate Care Unit
All inpatients 2015–2021		520
Diagnosis Anorexia nervosa F50.0	4121	410
Laboratory parameter complete	3755	339
Hospital stay > 28 d and BMI < 13		142
age	22.7 y, SD 9.8	26.4 y, SD 9.4
	(range 12–73 y)	(range 18–62 y)
female/male	3598/157	142 female
Weight (kg) admission	42.8 kg, SD 8.1	31.5, SD 3.6
	(range 22 -81 kg)	(23.5–40.1)
BMI admission	15.4 SD 2.4	11.5 SD 0.85
	(range 8.1–27)	(range 8.4–13)

**Table 2 nutrients-14-02378-t002:** Gender, BMI, and age (sample 1) gender.

BMI		N	Female	Age	BMI
<11	adolescent	3	2	15.0	10.7
	young adult (18 to 26 years)	28	28	21.1	10.5
	older adult (>26 years)	37	37	35.8	10.5
	all	68	67	10.50	10.5
11 to 12	adolescent	27	26	15.78	11.5
	young adult (18 to 26 years)	72	72	20.86	11.5
	older adult (>26 years)	51	50	35.71	11.5
	all	150	148	24.99	11.5
12 to 13	adolescent	90	90	15.58	12.6
	young adult (18 to 26 years)	104	102	20.33	12.5
	older adult (>26 years)	71	69	38.54	12.4
	all	265	261	23.59	12.5
13 to 14	adolescent	224	220	15.47	13.5
	young adult (18 to 26 years)	185	183	20.43	13.4
	older adult (>26 years)	108	102	38.39	13.4
	all	517	505	22.03	13.5
14 to 15	adolescent	286	277	15.37	14.5
	young adult (18 to 26 years)	202	191	20.42	14.5
	older adult (>26 years)	141	136	37.07	14.5
	all	629	604	21.86	14.5
15 to 16	adolescent	288	278	15.43	15.4
	young adult (18 to 26 years)	207	201	20.53	15.4
	older adult (>26 years)	136	128	36.65	15.4
	all	631	607	21.68	15.4
16 to 17	adolescent	221	211	15.63	16.4
	young adult (18 to 26 years)	183	176	21.28	16.4
	older adult (>26 years)	122	110	35.93	16.4
	all	526	497	22.31	16.4
17 to 18	adolescent	159	155	15.69	17.4
	young adult (18 to 26 years)	152	145	20.53	17.4
	older adult (>26 years)	108	93	35.46	17.4
	all	419	393	22.54	17.4
18 to 19	adolescent	80	78	16.11	18.4
	young adult (18 to 26 years)	97	92	20.84	18.4
	older adult (>26 years)	70	67	35.51	18.4
	all	247	237	23.47	18.4
19 to 20	adolescent	35	35	15.91	19.4
	young adult (18 to 26 years)	46	44	20.83	19.4
	older adult (>26 years)	45	39	36.78	19.4
	all	126	118	25.16	19.4
>20	adolescent	35	35	15.69	21.6
	young adult (18 to 26 years)	64	60	20.86	21.2
	older adult (>26 years)	52	40	35.65	22.1
	all	151	135	24.75	21.6
total	**adolescent**	**1448**	**1407**	**15.56**	**15.5**
	**young adult (18 to 26 years)**	**1352**	**1306**	**20.66**	**15.4**
	**older adult (>26 years)**	**955**	**885**	**36.61**	**15.5**
	**all**	**3755**	**3598**	**22.75**	**15.4**

## Data Availability

Due to the privacy policy of Schoen-Cinic Roseneck, the data cannot be made available online.

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
