# Peer review of "Liver Damage Is Related to the Degree of Being Underweight in Anorexia Nervosa and Improves Rapidly with Weight Gain"

_nutrients, 2022, doi:10.3390/nu14122378_

Round 1

Reviewer 1 Report

This is a clearly written manuscript which corroborates previous findings on liver damage-associated parameters in a larger number of patients of anorexia nervosa. I consider that it can be relevant for the audience to read and to have into consideration for future studies. I have no further comments.

Author Response

Thank you for your friendly comments. As there are no points to revise, there is no need for changes. 

Reviewer 2 Report

Paper  "Liver Damage Is Related To Degree Of Underweight In Anorexia Nervosa And Improves Rapidly With Weight Gain" consists of duty parts, which are necessary for structure of scientific work for Nutrients. The performance of paper is on good level and corresponds to demands, which are generally asked for scientific papers. Text is written on the understanding way - however, please read the text carefully and remove any minor imperfections and errors (for example line 136 - there are 2 parentheses in the text;  presenting results in tables and in the text - should be dots instead of commas).

Abstract and list of key words are performed in correct way. Appropriate methods were used for the statistical analysis and it was well performed. Tables are written in the understanding way. The data in the Figure are not legible. I suggest to improve the line thickness, enlarge the subtitles and improve the quality.

Author Response

Thak you for your comments and advices. We did the corrections and tried to make the figure more readable. We replaced, where ever necessary, the commas by dots.

Reviewer 3 Report

The present observational study investigates the relationship between the elevation of aminotransferases and malnutrition based on a considerable number of patients treated in the Roseneck Clinic within five years for anorexia nervosa.

Specifically, the study has two samples. The first sample consisted of 3755 outpatients, while the second was 142 inpatients admitted for malnutrition and followed for four weeks.

The Authors found that the increase in transaminases was very highly correlated with weight in sample 1. However, with renutrition in sample 2, transaminases decreased significantly within four weeks.

The sample is large, and the data has been correctly analyzed. Furthermore, the Authors discussed the pathophysiological hypothesis very well, in line with the data that emerged in their study.

Hence, the background, materials and methods, statistical analysis and results are promising. The graphs and tables are also well done. I report the inversion of the age and BMI values in table 2 for BMI <11. 

It can help Healthcare Professionals comprehend a disease still characterized by high mortality.

Finally, I point out some typos.

Author Response

Thank you very much for the comments and correction of typos, which we (hopefully) eliminated. The data for BMI < 11 had been mistaken and are now correct!